# Coherent characterisation of a single molecule in a photonic black box

Sebastien Boissier [1], Ross C. Schofield [1], Lin Jin[2], Anna Ovvyan[2], Salahuddin Nur [1], Frank H. L. Koppens[3,4], Costanza Toninelli [5], Wolfram H. P. Pernice [2], Kyle D. Major [1], E. A. Hinds [1] & Alex S. Clark [1✉]

Extinction spectroscopy is a powerful tool for demonstrating the coupling of a single quantum emitter to a photonic structure. However, it can be challenging in all but the simplest of geometries to deduce an accurate value of the coupling efficiency from the measured spectrum. Here we develop a theoretical framework to deduce the coupling efficiency from the measured transmission and reflection spectra without precise knowledge of the photonic environment. We then consider the case of a waveguide interrupted by a transverse cut in which an emitter is placed. We apply that theory to a silicon nitride waveguide interrupted by a gap filled with anthracene that is doped with dibenzoterrylene molecules. We describe the fabrication of these devices, and experimentally characterise the waveguide coupling of a single molecule in the gap.

[1] Centre for Cold Matter, Blackett Laboratory, Imperial College London, London, UK. [2] Physikalisches Institut, Westfälisches Wilhelms, Universität Münster, Münster, Germany. [3] ICFO—Institut de Ciencies Fotoniques, The Barcelona Institute of Science and Technology, Castelldefels (Barcelona), Spain. [4] ICREA—Institució Catalana de Recerça i Estudis Avancats, Barcelona, Spain. [5] LENS and CNR-INO, Sesto Fiorentino (FI), Italy. ✉email: alex.clark@imperial.ac.uk

Integrated photonic devices have allowed rapid progress to be made in applications, such as quantum sensing[1,2], quantum simulation[3] and quantum information processing[4]. However, the photon sources used in such devices are usually based on probabilistic nonlinear processes. A deterministic photon source would be more useful and single quantum emitters such as quantum dots[5], defect centres in crystalline materials[6] and single organic molecules[7] have shown great promise in this regard. Polycyclic aromatic hydrocarbons (PAH) were among the first solid-state quantum emitters to be studied[8,9], and have now become a significant alternative to other emitters[10,11]. Organic emitters have typically been coupled to inorganic photonic structures through evanescent coupling[10,12–14]. A single emitter coupled to an integrated photonic structure can act as a deterministic photon source and can also be used to build photon–photon interactions at the heart of a number of optical quantum computing schemes[15,16].

Typically, the coupling of a quantum emitter to a single-mode fibre or waveguide is quantified by carefully accounting for losses through all elements of the optical setup[17]. Another method is to compare the lifetimes of two similar emitters, one of which is not coupled to the photonic structure, and to use the Purcell effect to determine the coupling[18]. A third approach, known as extinction spectroscopy, relies on the interference between a continuous-wave laser and the resonance fluorescence of the emitter[16,19]. This interference affects the amplitude[20], phase[21] and photon statistics[22] of the transmitted and reflected fields. Exploration of this phenomenon has led to the demonstration of single emitters as optical transistors[23], phase switches[24] and quantum memories[25]. Extinction spectroscopy has been described in a number of settings, including in free space[26,27], with continuous waveguides[19] and with cavities[11,16,28].

Here, we expand the theory to describe an emitter placed in a photonic environment for which we cannot use modal decomposition to find a limited number of relevant modes. We only require that the environment be passive and linear and that the coupling of the emitter to the photonic reservoir be Markovian. We consider an arrangement where two guiding structures are used as input–output ports to the photonic structure and derive general results for the reflection and transmission spectra as a function of coupling efficiency. We apply this result to the characterisation of a single dibenzoterrylene (DBT) molecule coupled to a silicon nitride waveguide. The coupling is strongest at the maximum of the field and this motivates the geometry we consider here, where we investigate a waveguide structure interrupted by a microfluidic channel. We demonstrate that the channel can be filled at an elevated temperature by molten anthracene doped with DBT and that the DBT can have narrow resonances in the vicinity of the waveguide when cooled to cryogenic temperatures. We use extinction spectroscopy to characterise the coupling of the emitters to the waveguide, and because our photonic structure does not admit well-defined optical modes, we use our general theory to fit the transmission spectrum and quantify the coupling. Finally, we compare that measured coupling with the coupling expected from numerical simulations.

## Results

**Theoretical framework**. We consider the general system depicted in Fig. 1a, in which two optical guiding structures, labelled as the pump and probe waveguides, are connected by a photonic black box. We are interested in the coupling of the emitter to the pump and probe waveguides. In order to measure this, optical power $P_{in}$ is coupled into transverse mode $m$ of the pump guide, making a field $\mathrm{Re}\{\mathcal{E}_{in}\mathbf{u}_m(x,y)e^{i(kz-\omega t)}\}$. Here, $\mathbf{u}_m(x,y)$ gives the transverse

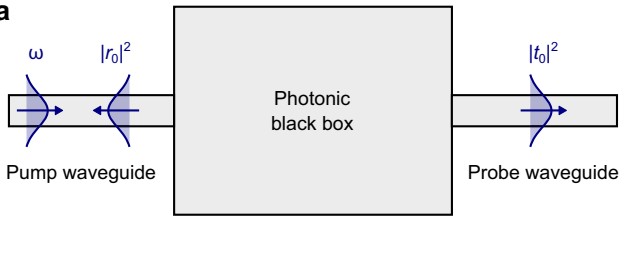

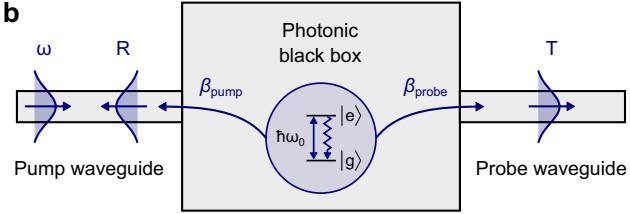

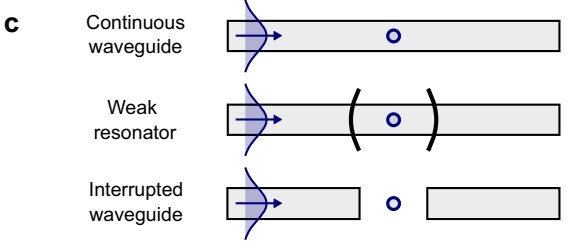

**Fig. 1 Schematic representation of the optical system to be characterised. a** Response of the system without the presence of a quantum emitter. Input light at frequency $\omega$ is transmitted (reflected) by the photonic structure with coefficient $|t_0|^2$ ($|r_0|^2$). **b** With the presence of a single emitter with ground state $|g\rangle$ and excited state $|e\rangle$ separated by energy $\hbar\omega_0$, the light interferes with the resonance fluorescence of the driven quantum system coupled to the pump and probe waveguides with efficiency $\beta_{pump}$ and $\beta_{probe}$, producing Fano transmission ($T$) and reflection ($R$) spectra. **c** Example Markovian structures covered by our general characterisation model, including a continuous waveguide, a resonator in the weak coupling regime and an interrupted waveguide.

distribution of the field in mode $m$, $z$ is the direction of propagation and $\mathcal{E}_{in}$ is the amplitude of the pump light in that mode. In the absence of the emitter, the transmitted pump light in mode $m$ of the probe guide is $\mathrm{Re}\{t_0\mathcal{E}_{in}\mathbf{u}_m(x,y)e^{i(kz-\omega t)}\}$, where $t_0$ is the complex transmission factor.

Consider an emitter with an upper level $|e\rangle$ and a lower level $|g\rangle$ (other levels are sufficiently far from resonance that they can be adiabatically eliminated from the coherent dynamics). When the emitter is put in place, the field has the option of being scattered by the emitter into the probe waveguide, as depicted in Fig. 1b. We make two key assumptions about the dynamics. First, we assume that the local density of electromagnetic states is constant over the spectral linewidth of the emitter[29]. This Markovian approximation is justified for most single-photon sources where a fast optical response is desirable. Second, we make the semi-classical assumption that the quantum correlations between the pump field and the emitter can be ignored[30]. With these assumptions, the total output field in the probe waveguide is given by[29]

$$\mathcal{E}_{out}(\mathbf{r}) = t_0\mathcal{E}_{in} + \mathcal{E}_{emit}^t\sigma^- , \qquad (1)$$

where we have dropped the factor $\mathbf{u}_m(x,y)e^{i(kz-\omega t)}$ from both sides of the equation. The operator $\sigma^- = |g\rangle\langle e|e^{i\omega t}$ ensures that the emission of a photon is accompanied by de-excitation of the emitter. With the emitter placed inside the structure, let the total

power scattered by the emitter at the frequency $\omega$ be $P_{\text{emit}}$, a fraction $\beta_{\text{probe}}$ of which is scattered into the probe guide mode $m$. Since the power in the guide is proportional to the square of the field, it follows that

$$\frac{|\mathcal{E}_{\text{emit}}^t|^2 \langle \sigma^+ \sigma^- \rangle}{|\mathcal{E}_{\text{in}}|^2} = \frac{\beta_{\text{probe}} P_{\text{emit}}}{P_{\text{in}}}, \tag{2}$$

where $\sigma^+ = e^{-i\omega t} |e\rangle \langle g|$ and the angle brackets indicate the steady-state expectation value of the atomic operator. With continuous-wave pumping in the near-resonant regime,

$$P_{\text{emit}} = \hbar \omega \gamma_1 \langle \sigma^+ \sigma^- \rangle, \tag{3}$$

where $\gamma_1$ is the population decay rate of the excited state due to radiation at the frequency $\omega$ of the pump light. This may be a partial decay rate because Raman sidebands and any non-radiative decay processes are not included here. In order to evaluate this, we need to know the field that drives the emitter. In "Methods", we show that this is related to $\beta_{\text{pump}}$ through the relation

$$\Omega^2 = 4\beta_{\text{pump}} \gamma_1 \frac{P_{\text{in}}}{\hbar \omega} . \tag{4}$$

Here, $\Omega$ is the Rabi frequency, defined as $\mathbf{d} \cdot \mathbf{E}(\mathbf{r}_0)/\hbar$, where $\mathbf{d}$ is the dipole transition matrix element and the pump field at the site of the emitter is $\text{Re}\{\mathbf{E}(\mathbf{r}_0)e^{-i\omega t}\}$. We choose $\Omega$ to be real without loss of generality.

On substituting Eq. (3) and Eq. (4) into Eq. (2), we find

$$|\mathcal{E}_{\text{emit}}^t| = \sqrt{4\beta_{\text{pump}} \beta_{\text{probe}}} \frac{\gamma_1}{\Omega} |\mathcal{E}_{\text{in}}|. \tag{5}$$

Hence, ignoring a global phase, the field at the output end of the guide is given by

$$\mathcal{E}_{\text{out}} = \left( |t_0| + \frac{\beta_{\text{eff}} \gamma_1}{\Omega} e^{i\phi_T} \sigma^- \right) |\mathcal{E}_{\text{in}}| . \tag{6}$$

Here, we have introduced $\phi_T$, which is the phase difference between the two transmitted fields due to propagation; a further phase shift will come from the lag of the dipole response $\sigma^-$. We have also defined $\beta_{\text{eff}} = \sqrt{4\beta_{\text{pump}} \beta_{\text{probe}}}$. Note that $\beta_{\text{pump}}$ and $\beta_{\text{probe}}$ are both between 0 and 1 but $\beta_{\text{pump}} \beta_{\text{probe}} \leq \beta_{\text{pump}}(1 - \beta_{\text{pump}}) \leq 1/4$, so the maximum value is $\beta_{\text{eff}} = 1$. It follows that the net transmission power is given by

$$\frac{P_{\text{out}}}{P_{\text{in}}} = \frac{\langle \mathcal{E}_{\text{out}} \mathcal{E}_{\text{out}}^\dagger \rangle}{|\mathcal{E}_{\text{in}}|^2}$$
$$= |t_0|^2 + 2|t_0| \frac{\beta_{\text{eff}} \gamma_1}{\Omega} \text{Re}\left( e^{-i\phi_T} \rho_{ge} \right) + \left( \frac{\beta_{\text{eff}} \gamma_1}{\Omega} \right)^2 \rho_{ee} , \tag{7}$$

where $\rho$ is the density matrix of the emitter with $\rho_{ge} = \langle \sigma^+ \rangle$, and $\rho_{ee} = \langle \sigma^+ \sigma^- \rangle$. These three terms correspond, respectively, to the transmitted pump power, the interference term between the pump field and the coherently scattered field and the scattered power, all in the probe guide.

The density matrix elements are found by solving the optical Bloch equations[19,29], with the result

$$\rho_{ee} = \frac{\frac{1}{2}S}{\left( \delta\omega/\Gamma_2 \right)^2 + 1 + S} ,$$
$$\rho_{ge} = -\frac{\Omega/(2\Gamma_2)}{\left( \delta\omega/\Gamma_2 \right)^2 + 1 + S} \left( \frac{\delta\omega}{\Gamma_2} + i \right) , \tag{8}$$

where $\delta\omega = \omega - \omega_0$ is the detuning of the laser from resonance, and $S = \frac{\Omega^2}{\Gamma_1 \Gamma_2}$ is the saturation parameter. Here, $\Gamma_1$ is the total decay rate of the upper-state population, while $\Gamma_2$ is the decay rate of the coherence $\rho_{ge}$ by all decoherence mechanisms. On substituting Eq. (8) into Eq. (7), we obtain the transmission

spectrum

$$\frac{P_{\text{out}}}{P_{\text{in}}} = |t_0|^2 - \left\{ 2\alpha\beta_{\text{eff}}|t_0| \left( \sin(\phi_T) + \frac{\delta\omega}{\Gamma_2} \cos(\phi_T) \right) - (\alpha\beta_{\text{eff}})^2 \right\} \frac{\Gamma_1/(2\Gamma_2)}{(\delta\omega/\Gamma_2)^2 + 1 + S} , \tag{9}$$

where $\alpha = \gamma_1/\Gamma_1$. A similar analysis gives the reflection spectrum

$$\frac{P_{\text{refl}}}{P_{\text{in}}} = |r_0|^2 - \left\{ 4\alpha\beta_{\text{pump}}|r_0| \left( \sin(\phi_R) + \frac{\delta\omega}{\Gamma_2} \cos(\phi_R) \right) - 4(\alpha\beta_{\text{pump}})^2 \right\} \frac{\Gamma_1/(2\Gamma_2)}{(\delta\omega/\Gamma_2)^2 + 1 + S} , \tag{10}$$

where $r_0$ is the reflection coefficient and $\phi_R$ is the reflection analogue of $\phi_T$.

In an experiment to measure the transmission as a function of frequency, Eq. (9) may be fitted to the spectrum. When $S \ll 1$ and the value of $\Gamma_1/(2\Gamma_2)$ is known, the fit will yield values for $|t_0|$, $\alpha\beta_{\text{eff}}$ and $\phi_T$. However, it is common in a real experiment for the light to be attenuated by the train of auxiliary optics so that the measured powers $\mathcal{P}_{\text{out}}$ and $\mathcal{P}_{\text{in}}$ have the ratio $\mathcal{P}_{\text{out}}/\mathcal{P}_{\text{in}} = \eta P_{\text{out}}/P_{\text{in}}$, and the value of $\eta$ is unknown. For large detuning, the measured transmission $\mathcal{P}_{\text{out}}/\mathcal{P}_{\text{in}}$ then takes the value $\eta|t_0|^2$. On normalising the data to this transmission, we have from Eq. (9)

$$T = \frac{\mathcal{P}_{\text{out}}}{\eta |t_0|^2 \mathcal{P}_{\text{in}}} = 1 - \frac{\alpha\beta_{\text{eff}}}{|t_0|} \left\{ 2\left( \sin(\phi_T) + \frac{\delta\omega}{\Gamma_2} \cos(\phi_T) \right) - \frac{\alpha\beta_{\text{eff}}}{|t_0|} \right\} \frac{\Gamma_1/(2\Gamma_2)}{(\delta\omega/\Gamma_2)^2 + 1 + S} . \tag{11}$$

In this case, the fit yields a value for $\alpha\beta_{\text{eff}}/|t_0|$, rather than $\alpha\beta_{\text{eff}}$. One may determine $|t_0|$ by an auxiliary experiment that compares the device with another that contains no emitter and has $|t_0| = 1$. Eq. (10) can be modified in an exactly similar way to give the reflection spectrum as a function of $\alpha\beta_{\text{pump}}/|r_0|$, while $|r_0|$ is determined by comparison with a reflector having $|r_0| = 1$. Finally, the value of $\beta_{\text{probe}}$ can be deduced from the combination of $\beta_{\text{eff}}$ and $\beta_{\text{pump}}$. Of course, $\beta_{\text{probe}}$ could also be determined from a reflection measurement made on the output side.

For structures where a normal mode decomposition is appropriate, for example, a continuous waveguide or weak cavity as depicted in Fig. 1c, most of the parameters in Eq. (9) and Eq. (10) can be calculated analytically, as we consider further in Section I of the Supplementary Information. Our method becomes essential when dealing with structures that cannot be described by a simple mode decomposition, such as the waveguide gap depicted in Fig. 1c.

**Microfluidic integration of single molecules with waveguides.** In the early days of single-emitter spectroscopy, it was found that large PAH molecules such as pentacene[8], terrylene[31], diben-zanthanthrene (DBATT)[32] and dibenzoterrylene (DBT)[33] could be hosted in PAH crystals to form stable quantum emitters in the solid state. In this work, we use DBT-doped anthracene. The molecular structures are shown in Fig. 2a and relevant energy levels of DBT are drawn in Fig. 2b. This well-studied combination has a very weak singlet–triplet inter-system crossing, is highly photostable, has a high probability of radiative decay on the zero-phonon line (ZPL) (shown in blue)[33,34] and has a lifetime-limited resonance width at cryogenic temperatures[35].

The coupling of photons to single PAH molecules has been used in bulk material to demonstrate, for example, a single-molecule optical transistor[23] and few-photon nonlinear optics[36], but for applications such as a deterministic photon source, stronger coupling is desirable. A natural way to achieve that is to integrate the emitters into a photonic structure[10,12,13,15], and it is convenient to grow doped organic crystals around the structure by solidifying from a molten mixture[12,37]. Normally, the structure

 3

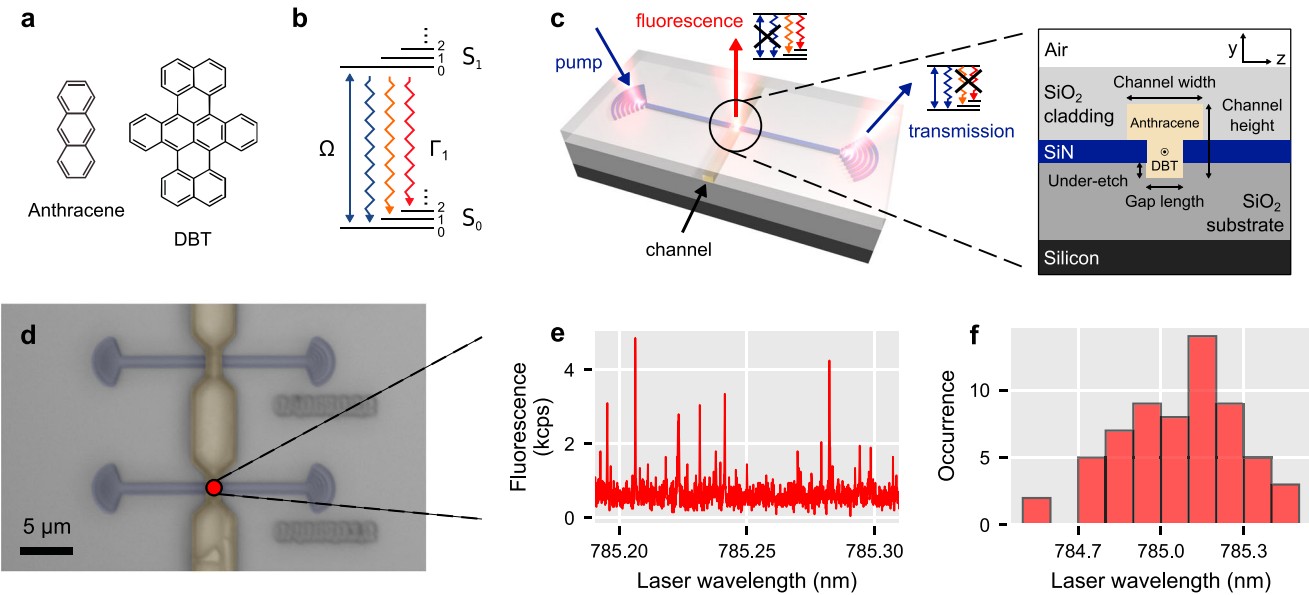

**Fig. 2 Localised growth of DBT-doped anthracene in the vicinity of an interrupted waveguide. a** Molecular structure of anthracene and dibenzoterrylene (DBT). **b** Jablonski diagram of DBT ground ($S_0$) and excited ($S_1$) states excited at Rabi frequency $\Omega$ and decaying with rate $\Gamma_1$. Triplet states are ignored as the inter-system crossing is very weak[33]. **c** Overview of the grating couplers, the interrupted silicon nitride (SiN) waveguide and the microfluidic channel crossing it. The zoom-in shows details of the intersection between guide and channel. **d** False-colour optical-microscope image of two devices with the microfluidic channels filled. **e** Fluorescence excitation spectrum of molecules near the gap in a device at cryogenic temperature. The molecules are excited from the "pump" waveguide and fluorescence is collected by the confocal microscope from the red dot shown in **d**. **f** Wavelength distribution of the DBT resonances from the same confocal spot.

is made of inorganic material and the organic molecule couples to an evanescent field. However, the molecule is usually unstable at less than a few hundred nanometres from the inorganic/organic interface[38], and therefore it can only be placed in the tail of the evanescent field where the dipolar coupling to the photonic mode is weak. Here, we take a different approach, shown in Fig. 2c, where a silicon nitride waveguide having grating couplers at each end is interrupted by a sub-wavelength gap. After fabricating the waveguide chip, molten anthracene doped with DBT is drawn by capillary forces along a microfluidic channel that cuts across the waveguide and fills the gap, as depicted in Fig. 2c. A numerical simulation, details of which are given in Section II and Supplementary Fig. S1 of the Supplementary Information, shows that the coupling efficiency $\beta_{\mathrm{eff}}$ for a molecule sitting at the centre of the gap decreases rapidly with the length of the gap. However, with a gap of 400 nm, this can be as high as 30%. A smaller gap can yield higher coupling, but the guide faces on each end of the gap are then close enough to the molecule that they may compromise its optical properties. The coupling can, of course, be much higher with the introduction of a cavity[10].

We began device fabrication with a silicon wafer that had a layer of thermal oxide covered by silicon nitride, and we patterned the interrupted waveguides and grating couplers in the silicon nitride. The microfluidic channels were then fabricated from a sacrificial resist layer on top of which $SiO_2$ was sputtered. We cleaved the chips to expose the channel entrance at the facets and baked the sample to remove the resist, thereby opening hollow channels. Finally, we filled the channels with DBT-doped crystalline anthracene by controlled heating and subsequent cooling. See "Methods" for details of device fabrication and filling of the capillaries.

To verify that we had stable emitters in the vicinity of the waveguide, we performed fluorescence spectroscopy at a cryogenic temperature under a microscope (see "Methods" for a detailed description of the optical setup). A waveguide chip was filled with DBT-doped anthracene at $10^{-4}$ molar fraction, then

cooled in the cryostat to 4.7 K and positioned so that a device having 400-nm gap length and 1-$\mu$m channel width (see Fig. 2c) was in focus at the centre of the field of view. Figure 2d shows a false-colour white-light image of the structure. A cw laser was focused onto one of the grating couplers to excite molecules from the "pump" side of the waveguide, and was continuously scanned at low power between 784.5 nm and 785.5 nm to cover the inhomogeneous width of the $S_{0,0} \leftrightarrow S_{1,0}$ transition. The light was then collected from the vicinity of the waveguide gap and sent to a photon counter. An 800-nm long-pass filter removed any scattered laser light together with the ZPL fluorescence, leaving only the red-shifted fluorescence. We plot a slice of the scan in Fig. 2e, which reveals the characteristic Lorentzian resonance peaks of many DBT molecules having a range of resonant frequencies, each being shifted according to its local environment. The histogram of these frequencies shown in Fig. 2f corresponds to a spectral density of 0.2 molecules per GHz at 785 nm, so the individual molecular lines are well resolved. Some of the light is collected from molecules that are well away from the gap but are excited by scattered pump light, and these have poor coupling to the guide. To characterise the strength of the coupling, we therefore use the extinction spectroscopy method developed above, as described next.

**Characterisation of the coupling**. In this section, we make a practical demonstration of our theory through a transmission measurement, which we compare with Eq. (9). We collected light from the grating on the "probe side" of the waveguide and used a $785 \pm 3$-nm band-pass filter to remove the red-shifted fluorescence and most of the local phonon sideband[39]. (We show in Section IV of the Supplementary Information that imperfect filtering has a negligible effect). The grating selectively couples to the $x$-polarised waveguide mode, which is the mode that we pump. We scanned the pump frequency over the resonance of a single DBT molecule and recorded both the red-shifted fluorescence from the gap and the resonant transmission from the

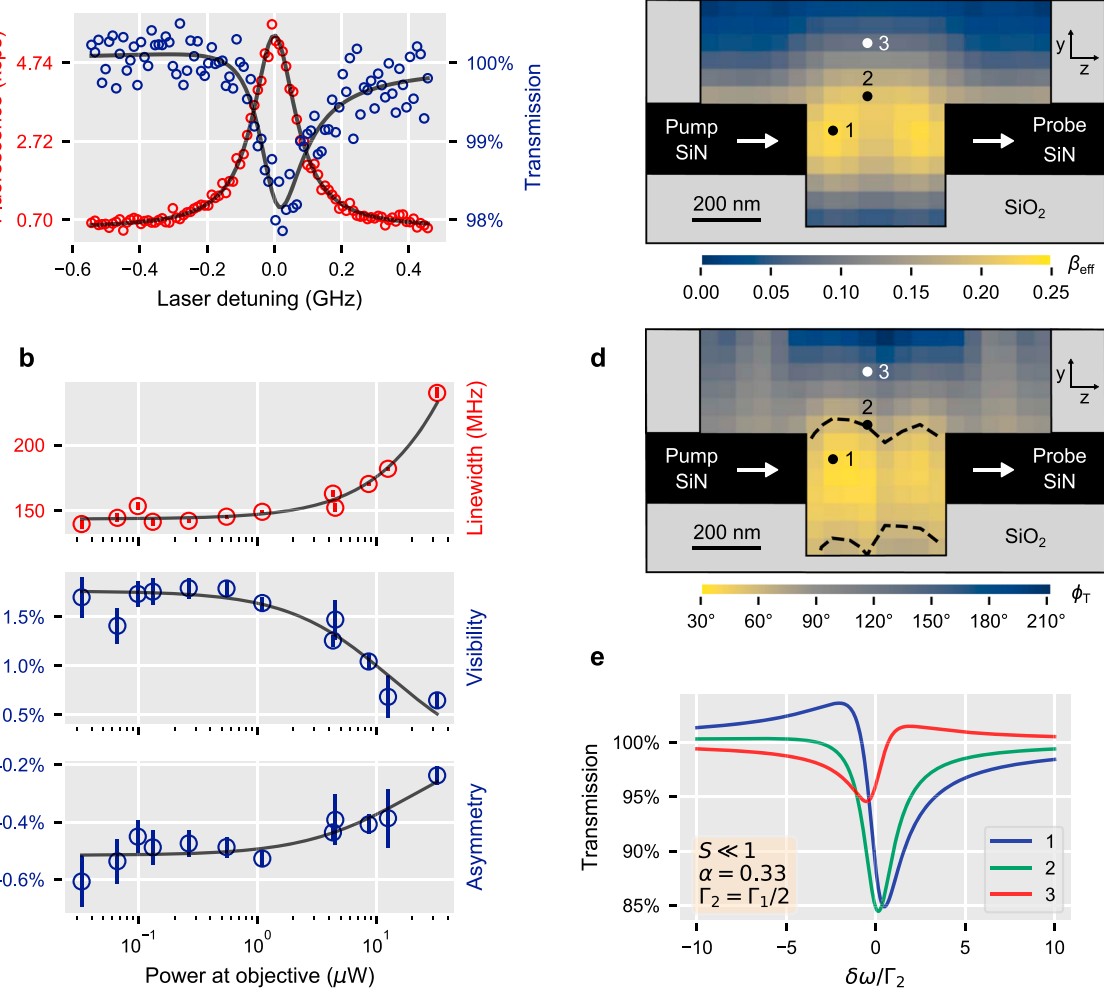

**Fig. 3 Coherent characterisation of a coupled single molecule by extinction spectroscopy. a** Experimental data. Red circles: red-shifted fluorescence emerging from the gap, fitted with a Lorentzian curve. Blue circles: transmission at 785 nm observed through the output grating, normalised and fitted according to Eq. (11). **b** Plots of full width at half maximum (FWHM) linewidth, visibility (*V*) and asymmetry (*q*) as a function of the pump power. Error bars signify one standard deviation from fitting the transmission at each power. **c**, **d** Computation of the coupling efficiency $\beta_{\text{eff}}$ and propagation phase difference $\phi_T$ for a dipole along the *x* direction, placed in the mirror symmetry plane of the structure. **e** Normalised transmission spectrum from Eq. (11) expected for lifetime-limited DBT molecules placed inside the channel, at positions 1–3 shown in **c** and **d**.

output grating, as indicated in Fig. 2c. The data are plotted in Fig. 3a as open circles in red and blue for the fluorescence and transmission, respectively. A solid line shows the least-squares fit of a Lorentzian to the fluorescence data, which gives us the linewidth, $2\Gamma_2\sqrt{1+S}$ (full width at half maximum (FWHM)). Knowing that, we then fit Eq. (11) to the transmission data to produce the solid line through the blue data points. For this second fit, we express Eq. (11) as the Fano lineshape

$$T(\epsilon) = \frac{1 - (V + q^2)}{1 + \epsilon^2} + \frac{(q + \epsilon)^2}{1 + \epsilon^2}, \qquad (12)$$

with

$$V = \beta\big(2\sin(\phi_T) - \beta\big)\frac{\Gamma_1/(2\Gamma_2)}{1 + S}, \qquad (13)$$

$$q = -\beta\cos(\phi_T)\frac{\Gamma_1/(2\Gamma_2)}{\sqrt{1 + S}}, \qquad (14)$$

where $\epsilon = \delta\omega/(\Gamma_2\sqrt{1+S})$ is the normalised detuning and $\beta = \alpha\beta_{\text{eff}}/|t_0|$ is the scaled coupling efficiency. This fit gives the values of *V* and *q*. Repeated scans at twelve different pump powers gave us values for the FWHM linewidths, visibilities (*V*)

and asymmetries (*q*) that are plotted in Fig. 3b. On extrapolating to the limit of low power, we find the values $\Gamma_2/\pi = 144(2)$ MHz, $V_0 = 1.8(1)\%$ and $q_0 = -0.52(1)\%$. We note that the value of $\Gamma_2/\pi$ is significantly greater than the ~ 35-MHz natural linewidth because our cryostat only cooled the sample down to ~4.7 K, whereas the minimum width is reached at ~3.5 K. Eq. (13) and Eq. (14) give two solutions for $\beta$ and $\phi_T$. In the limit of small *S*,

$$\beta_\pm = \sqrt{2 - \tilde{V}_0 \pm 2\sqrt{1 - \tilde{q}_0^2 - \tilde{V}_0}}, \qquad (15)$$

$$\phi_{T\pm} = \text{atan2}\big(4\tilde{q}_0^2 - \tilde{V}_0(\tilde{V}_0 + \beta_\pm^2 - 4), \\ 2\tilde{q}_0(2\tilde{V}_0 + \beta_\pm^2 - 4)\big), \qquad (16)$$

where $\tilde{q}_0 = q_0/(\Gamma_1/(2\Gamma_2))$ and $\tilde{V}_0 = V_0/(\Gamma_1/(2\Gamma_2))$. We use the function atan2( numerator, denominator ) to ensure that $\phi_T$ is placed in the correct quadrant.

In order to derive $\beta_{\text{eff}}$ from $V_0$ and $q_0$, we measured $|t_0|$ by comparing the off-resonant transmission of the device with the transmission of a second device, which was identical, except that the waveguide had no gap. We scanned the laser frequency to look for possible cavity resonances in the optical setup, which

would have invalidated the method but found only a very weak modulation. This comparison gave $|t_0| = 0.63(6)$, which differs slightly from the numerically calculated transmission $|t_{sim}| = 0.81$, perhaps because our simulation simplifies the anisotropic refractive index of the anthracene. Having measured $|t_0|$, the only external parameters needed to deduce a value for $\beta_{eff}$ are $\Gamma_1^{-1} = (4.5 \pm 1)\,\text{ns}$[40] and $\alpha = 0.33$[35,39], both known from bulk measurements of DBT in anthracene. We have checked (see Supplementary Information Fig. S3d) that $\Gamma_1$ is not appreciably altered when the anthracene channel is narrow, and we have shown (see Supplementary Information Fig. S1a) that the radiation rate $\gamma_1$ and the branching ratio $\alpha$ are not significantly altered by the electromagnetic response of the waveguide gap. With these two inputs, we find that the $\beta_+$ solution gives the unphysical result $\beta_{eff} > 1$, so we conclude that $\beta_{eff} = \beta_- |t_0|/\alpha = 9(2)\%$, with the error bar coming roughly equally from the uncertainty in $|t_0|$ and from the other uncertainties combined. The corresponding solution for the phase difference is $\phi_T = 61(2)°$ (independent of $|t_0|$), with the error bar coming primarily from the uncertainties in $q_0$ and $V_0$.

It is instructive to compare these results for $\beta_{eff}$ and $\phi_T$ with a numerical simulation (see "Methods"). Figure 3c shows $\beta_{eff}$ for a dipole transverse to the guide (along $x$), placed in the $yz$ plane centred on the guide. (The coupling at the centre of the 400-nm gap is less than the maximum possible 30% because the height and width of the guide are not perfectly optimised). The coupling is the strongest for an emitter placed in the gap, but we note that an emitter outside the gap and close to the guide couples to the evanescent field, as seen by the yellow strip running along the outside of the guide. Figure 3d shows the propagation-phase difference $\phi_T$. This phase varies strongly with the position in the gap, in contrast to the behaviour when coupling to a cavity. Also, we find that $\phi_T$ tends to 90° when the emitter couples to the evanescent field on the side of the guide and far from the gap, as expected for coupling to a continuous waveguide. See Section I of the Supplementary Information for treatments of the continuous waveguide and weak cavity cases. In Fig. 3e, we plot the transmission spectra calculated for weakly pumped, ideally polarised DBT molecules at each of the three positions marked in Fig. 3c, d.

The dashed lines in Fig. 3d show where a dipole lying in the $yz$ plane through the centre of the guide would give the measured value $\phi_T = 61°$. If the molecule is in this plane, we expect it to be near the upper contour, for example in the position marked 2, because the lower one is too close to the substrate for photostability. On this line, the calculated coupling efficiency varies in the range $20-21\%$, which is to be compared with the $9(2)\%$ we have measured. Our molecule has no reason to be aligned along $x$, so the simulation would be consistent with our measurement if the molecule makes an angle of $\theta = 49°$ to the $x$ axis. Of course, there is also no reason for the molecule to sit in the plane $x = 0$. Looking at the whole surface where $\phi_T = 61°$, we find that the simulated coupling varies in the range $11-21\%$, and conclude therefore that $\theta$ is in the range $25°-49°$.

## Discussion

We have demonstrated how to characterise the coherent scattering of light by a single quantum emitter, in a photonic environment that cannot be decomposed into a small number of relevant modes. We have shown that the transmission and reflection spectra are described by Fano lineshapes, from which one can extract the coupling efficiencies without needing precise knowledge of the photonic structure. Our method generalises extinction spectroscopy to complex geometries, yielding values for coupling efficiency without needing to measure in detail all

the losses in the system. Further, the propagation phase shift $\phi_T$ can provide some information on the position of the emitter within the structure and on the orientation of its transition dipole.

We have also demonstrated a method to integrate a single molecule into photonic structures on a chip by using microfluidic channels to bring doped crystals to the desired locations. In this initial demonstration, we have used the channels to place molecules in a simple gap, for which the value of $\beta_{eff}$ is unlikely to exceed 50%, even with high refractive index waveguides made from titanium dioxide or gallium phosphide. However, we plan to use the same method to place molecules conveniently into more complex environments, such as slotted waveguides[15,41] or resonator structures[42], for which $\beta_{eff}$ can reach close to 100%[18]. This work also opens the possibility of integrating molecular quantum emitters with photonic components such as beam splitters, interferometers and detectors, to study quantum networks and integrated quantum sensors[43]. In addition, we have shown that anthracene crystals can be highly doped to achieve densities on a chip of hundreds of emitters per $\lambda^3$ per nm. This could enable the study of collective behaviour of coupled quantum systems such as polaritonic light–matter states[19] or direct dipole–dipole interactions[44].

## Methods

**Derivation of Eq. (4)**. The classical pump field $\{\mathbf{E}_m^f, \mathbf{H}_m^f\}$ propagates forward (towards the black box) in transverse mode $m$ at frequency $\omega$ and with power $P_{in}$. This field leaves the guide and enters the black box, where it induces a dipole moment $\mathbf{D}$ in an emitter that radiates the field $\{\mathbf{E_d}, \mathbf{H_d}\}$ with power $P_d$. A fraction $\beta_{pump}$ of that radiated power goes back into the pump guide. From the orthogonality of modes[45] we have

$$\beta_{pump} = \frac{\left|\frac{1}{4}\int\left(\mathbf{E_d}\times\left(\mathbf{H}_m^b\right)^* + \left(\mathbf{E}_m^b\right)^*\times\mathbf{H_d}\right)\cdot d\mathbf{S}\right|^2}{P_{in} P_d}, \quad (17)$$

where the superscript $b$ denotes the mode propagating backwards (away from the black box).

Wanting to relate these fields to $\mathbf{D}$, we note that the dipole at position $\mathbf{r}_0$ produces a current density $\mathbf{j_d}(\mathbf{r}) = -i\omega\mathbf{D}\delta(\mathbf{r} - \mathbf{r}_0)$. Similarly, the pump field may be viewed as the result of (fictitious) electric and magnetic current densities $\mathbf{j}_{in}(\mathbf{r}) = \boldsymbol{\delta n}\times\mathbf{H}_m^f$ and $\mathbf{m}_{in}(\mathbf{r}) = \boldsymbol{\delta n}\times\mathbf{E}_m^f$[45]. These lie on a plane surface $S$ far from the black box, whose normal is parallel to the direction of propagation, and $\boldsymbol{\delta n}$ is a Dirac delta function along the normal. Now we can make use of the reciprocity theorem[46] to write

$$\int\mathbf{j_d}\cdot\mathbf{E}\,dV = \int(\mathbf{j}_{in}\cdot\mathbf{E_d} + \mathbf{m}_{in}\cdot\mathbf{H_d})\,dV, \quad (18)$$

where $\mathbf{E}$ is the pump field and the integrals are over an arbitrarily large volume that includes $S$. On evaluating these integrals with the explicit current densities we find that

$$-i\,\omega\,\mathbf{D}\cdot\mathbf{E}(\mathbf{r}_0) = \int\left(\mathbf{E_d}\times\left(\mathbf{H}_m^b\right)^* + \left(\mathbf{E}_m^b\right)^*\times\mathbf{H_d}\right)\cdot d\mathbf{S}, \quad (19)$$

where we have used the relations $\mathbf{E}_m^f = \left(\mathbf{E}_m^b\right)^*$ and $\mathbf{H}_m^f = -\left(\mathbf{H}_m^b\right)^*$. Using Eq. (19) to eliminate the integral in Eq. (17) we have

$$\beta_{pump} = \frac{1}{16}\frac{\omega^2|\mathbf{D}\cdot\mathbf{E}(\mathbf{r}_0)|^2}{P_{in} P_d}. \quad (20)$$

Connecting the classical dipole to the quantum emitter, we replace the ratio $|\mathbf{D}\cdot\mathbf{E}(\mathbf{r}_0)|^2/P_d$ by $(2\hbar\Omega)^2/(\hbar\omega\gamma_1)$[46]. Both $\gamma_1$ and the Rabi frequency $\Omega$ are defined in the main text. With this substitution in Eq. (20), we obtain the result given in Eq. (4).

**Device fabrication**. The waveguides are fabricated from a 200-nm-thick silicon nitride layer on 2 μm of silica on silicon. The waveguide patterns are first written into ma-N 2403 resist by electron beam lithography and transferred into the underlying silicon nitride layer by reactive ion etching with a $CHF_3$ plasma. We overetch the silicon nitride by 150 nm so that the middle of the waveguide sits 250 nm away from the bottom surface. In this way, the position of maximum coupling is not too close to the bottom surface. The waveguides on the chip have a width of 400 nm and gap lengths ranging from no gap to 400 nm. We terminate the waveguides with gratings based on concentric circles. To avoid reflections, the gratings are designed to couple light at an angle of 10° to the vertical.

To overlay the microfluidic channels, we first spin-coat a 1-μm layer of AZ nLOF 2020 resist that is diluted 4:1 (resist:solvent w/w) with PGMEA. Electron beam lithography exposes the resist along channels that are perpendicular to the

waveguides and aligned with the gaps. We then deposit 2 μm of SiO$_2$ on top of the resist using RF sputtering. Next, the sample is cleaved to expose the resist channels on both facets. Finally, we place the sample in a furnace that is heated to 550 °C in the ambient atmosphere. Under these conditions, we find that the resist is released from the channels without leaving any residue, and we are left with open structures that can be filled with molten DBT-doped anthracene.

**Capillary filling**. In order to fill the microfluidic channels with doped anthracene, we use growth from the melt by solidification[12,37]. We first place a small quantity of DBT-doped anthracene powder ($10^{-4}$ mol/mol concentration) on the facets of the chip. The sample is then put on a hotplate in a glove box that is continuously purged with nitrogen. We heat the sample at a rate of $5\,°C\,s^{-1}$ and hold the temperature at 210 °C until the channels are visibly filled by the melted material. Finally, we cool the sample at a rate of $-5\,°C\,s^{-1}$ causing the anthracene to crystallise. This yields long stretches of the capillaries filled by solid anthracene. We check the quality of the DBT molecules in the capillaries using cryogenic fluorescence spectroscopy and we show in Section 3 and Supplementary Fig. S3 of the Supplementary Information that the spectral stability is not appreciably affected by the constrained geometry of the microfluidic channel.

**Optical setup**. The optical apparatus was a three-beam confocal microscope built around a closed-cycle cryostat (Cryostation, Montana Instruments), as illustrated in Supplementary Fig. S2 of the Supplementary Information. The primary excitation light came from continuously tunable titanium:sapphire laser (SolsTiS, MSquared) that was power-stabilised using an acousto-optic modulator and a proportional integrated-derivative controller (SIM960, SRS). The light was delivered to the apparatus through a single-mode fibre, then collimated with an aspheric lens and polarised before passing through a half-wave plate and a band-pass filter (F1) to produce a linearly polarised beam with adjustable polarisation angle and spectral purity. This entered a 10% transmission (90% reflection) beamsplitter (BS), and the transmitted light was sent to a pair of electronically controlled galvanometer mirrors (GM). Through the use of two lenses in a "4f" configuration (L1, L2), the angular change in the galvanometer mirrors allowed us to adjust the angle of incidence onto an objective lens (LD EC Epiplan-Neofluor ×100, 0.75NA, Zeiss) inside the cryostat without translating across the objective aperture. This in turn caused a focused spot to be raster-scanned across the sample. The back aperture of the objective was overfilled to ensure the minimum spot size of 720-nm full-width half-maximum. The sample was mounted on a 3-axis piezo-controlled translation stage (PS, Attocube), which we used to locate waveguides and bring them into focus. Molecule fluorescence followed the beam path back to the 90:10 BS where the 90% reflected portion passed through a long-pass filter (F2) to remove the excitation laser before being collected in multimode fibre and detected on a silicon avalanche photodiode. By inserting a pellicle BS into the excitation path after the scanning mirrors, we introduced white light (WL) from a lamp onto the sample. This light was then reflected from the sample and off another pellicle BS above the cryostat to an electron-multiplying charge-coupled device (CCD) camera (iXon, Andor), which took wide-field images, such as that shown in Fig. 2d. A second single-mode fibre input (shown within the rectangle labelled "Grating Coupling") was collimated, polarised, filtered and steered onto a (90:10) beam splitter, before being combined with the main beam path in the "4f" lens setup using a 50:50 beam splitter. The steering mirrors allowed the beam to couple into the pump guide through its grating coupler, giving a typical total coupling efficiency of 8% from fibre to waveguide. Light emerging from the probe guide grating coupler was directed back to a final single-mode fibre (in the rectangle) and thence to the detector that recorded the transmission spectrum.

**Finite-difference time-domain simulations**. The numerical simulations of the device are performed with three-dimensional finite-difference time-domain (FDTD) analysis using the Meep software package[47]. The structural parameters, as defined in Fig. 2c, are waveguide width = 400 nm, waveguide height = 200 nm, gap length = 400 nm, under-etch = 150 nm, channel width = 1 μm and channel height = 1 μm. We use a mesh size of 16 nm and perfectly matched layers to simulate open boundaries. Anthracene is a biaxial material but for simplicity, we choose to approximate it as isotropic with refractive index $n = 1.8$.

To compute the transmission through the gap, we use a continuous eigensource to excite the $x$-polarised mode of the pump waveguide. We determine the power transmitted into the $x$-polarised mode of the probe waveguide by projecting the field at the output end onto that mode. For coupling efficiency calculations, we use a continuous dipole source placed at a given position in the channel and monitor the total power emitted together with the power coupled into the $x$-polarised modes of the waveguides.

FDTD simulations also allow us to calculate the phase difference $\phi_T$. Using a continuous eigensource to excite the $x$-polarised mode $m$ of the pump waveguide, we first compute the phase shift of the transmitted light, $\text{Arg}(t_0)$, which is the phase difference between light in mode $m$ at the entrance of the probe guide and the exit of the pump guide. Mode decomposition is used to isolate the field coupled to mode $m$ of the probe waveguide. For the propagation phase shift of the scattered light, we place an electric dipole at the position of the molecule. The dipole oscillates in phase with the pump field at that position, but the pump field is not

turned on. Again we take the difference between the phase of the (dipole) field in mode $m$ at the entrance to the probe guide and that of the pump field (if it were turned on) at the exit of the pump guide. Calling this latter phase shift $\Delta\phi$, we have $\phi_T = \Delta\phi - \text{Arg}(t_0)$.

## Data availability

The data presented here can be accessed from Zenodo at https://doi.org/10.5281/zenodo.4247265 and used under the Creative Commons CCZero license.

## Code availability

The code associated with this paper is available from the corresponding author upon reasonable request.

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

## Acknowledgements

We thank Jon Dyne and Dave Pitman for their expert mechanical workshop support. We also thank David Mack and Javier Cambiasso for their help with nanofabrication. This work was supported by EPSRC (EP/P030130/1, EP/P01058X/1, EP/R044031/1, EP/P510257/1 and EP/L016524/1), the Royal Society (UF160475, RGF/R1/180066 and RGF/EA/180203) and the EraNET Cofund Initiative QuantERA under the European Union's Horizon 2020 research and innovation programme, Grant No. 731473 (ORQUID Project).

## Author contributions

S.B. and E.A.H. formulated the theory; S.B. performed the numerical simulations; S.B., S.N., L.J., A.O. and W.H.P.P. designed and fabricated the nanophotonic devices; S.B., R.C.S., K.D.M. and A.S.C. built the experiment and took the data; S.B., R.C.S., K.D.M., E.A.H. and A.S.C. analysed the data. S.B., R.C.S., L.J., A.O., S.N., F.H.L.K., C.T., W.H.P.P., K.D.M., E.A.H. and A.S.C. discussed the results. S.B. and E.A.H. wrote the initial draft paper, and S.B., R.C.S., L.J., A.O., S.N., F.H.L.K., C.T., W.H.P.P., K.D.M., E.A.H. and A.S.C. contributed to the final paper. E.A.H. and A.S.C. led the project.

## Competing interests

The authors declare no competing interests.
