## [Peer Review File · Nature Communications]

Reviewers' Comments:

Reviewer #1:

Remarks to the Author:

The authors have developed theory in the context of coupling a single photon emitter to a photonic structure. They have developed a theoretical method to deduce the coupling efficiency using only reflection and transmission spectra. They have then applied this theory to an experimental setup in which a DBT molecule is coupled to a SiN waveguide. The broader context for this work is the hot topic of genuine single photon sources integrated with waveguides for quantum information processing.

As the work stands, it chiefly demonstrates that coupling can be inferred from spectra without knowledge of the photonic structure. Their experimental results are consistent with this. The work is perhaps less convincing on the efficacy of their methods as a useful approach to integrate DBT molecules, and I'm not sure whether the field has been advocated in this regard - perhaps the authors could add more of a conclusive statement here.

Reviewer #2:

Remarks to the Author:

Please see the attached pdf file.

The manuscript addresses the problem of extracting the coupling efficiency for a single quantum emitter and a waveguide. The emitter in this study was a single dibenzoterrylene (DBT) molecule. The coupling efficiency is the probability for an emitted photon to go into the waveguide mode. The extraction is performed by fitting experimentally measured transmission curves with analytically obtained functions that contain several unknown parameters, including the coupling efficiency. The experimental transmission shows a clear Fano-type line, in agreement with the theoretical expectation.

In general, the subject of the manuscript is quite interesting and is well aligned with the current efforts directed to the practical realization of quantum photonic devices. The manuscript is well written and the results are presented adequately.

While the results seem to be reasonable I do have some concerns about their analysis. I list my comments below. Based on this I feel that the manuscript requires further clarifications and revisions before making any decisions on its publication.

1. The extraction procedure is based on the assumption that the radiative decay rate γ_1 , first defined in Eq. (3), remains unchanged when the molecule is put inside the waveguide. The parameter γ_1 is set using $\alpha = \gamma_1/\Gamma_1 = 0.33$ based on some measurements for DBT molecules in anthracene from the literature. However, it seems to me that the presence of a waveguide should change the radiative decay rate. The changes may be less significant compared to that for a resonator but still large enough to affect the extraction results. Furthermore, the extraction procedure uses $\Gamma_1^{-1} = (4.5 \pm 1)$ ns. Does this value remain unchanged when the molecules are inserted into the waveguide? The assumptions should be stated more clearly and justified. Furthermore, it makes sense to list in one place all external parameters which are required for the proposed extraction procedure.
2. The extraction procedure uses measurements at several values of the pump power. Is it really necessary? Can one use only weak power to remain in the linear regime? Is there any advantage of going into the nonlinear regime besides being able to verify the power dependence of spectra, which does not seem to be useful for the extraction of the linear parameters.
3. The approach yields only some effective coupling coefficient β_{eff} . However, the goal set in the manuscript, line 70, is to find β_{probe} and β_{pump} . Can the two parameters be obtained directly from the experiments? Does one have to rely further on some geometrical symmetry to obtain β_{probe} and β_{pump} from β_{eff} ?
4. Page 5, line 268: scanning was performed “over the resonance of a single DBT molecule.” Does that mean that all molecules have non-overlapping spectral lines? Since all molecules seem to be identical, what is the physical mechanism that makes their emission lines to be separated? What is the chance that the lines of several molecules overlap?
5. The extraction is presented only in one particular case, i.e., for a specific molecule. It would certainly be useful to prove the results by applying the procedure to the spectra of several molecules.
6. Page 2, line 89: It says that “the optical reservoir decays faster than all other relevant time scales”. It would be useful to list explicitly the relevant time scales.
7. As far as I understand, while the paper proposes an approach to measure the coupling efficiency it does not specify the location of the molecule for which the efficiency is measured. Is that correct?
8. Figure 3b shows both visibility and asymmetry in percent. However, the numbers quoted in the text use percent only for the visibility parameter ($V_0 = 1.8(1)\%$) but not for the asymmetry parameter ($q_0 = -5.2(1) \times 10^{-3}$). It is better to be more consistent.
9. Page 3, line 128: it says “We have also replaced $\sqrt{4\beta_{\text{pump}}\beta_{\text{probe}}}$ by β_{eff} ”. Why not just write clearly that $\beta_{\text{eff}} = \sqrt{4\beta_{\text{pump}}\beta_{\text{probe}}}$?
10. Figs. 3(c,d): the colorbars require some marks and values at the ends.
11. Page 6, line 333: remove an extra *of*.
12. Page 1, line 18: typo in *through*

Reviewer #3:

Remarks to the Author:

Report on Boissier et al. "Coherent characterisation of a single molecule in a photonic black box"

This report concerns the generation of single photons, with a view to applications in quantum information, networks and simulation. The report covers experimental and theoretical aspects, and provides some very nice results. I have a number of comments on detail, points that I imagine the authors will be able to deal with quite easily. They are listed below. Before making those comments I wish to discuss my view about how important this work may be.

First, let me be clear that my own expertise is not a perfect overlap with that of this work. My own research is in nanophotonics, where I do have considerable experience with spontaneous emission into nanostructures.

The experimental work is clear, well executed and interesting. The theoretical aspects are devoted primarily to the problem of how one deals with situations – as presented here – in which the photons produced by the emitter couple to an unknown medley of optical modes (rather than for example a single waveguide mode). It is this novelty/value of this theoretical/conceptual aspect that I am not clear about. If I make a single photon source – such as that used here, or in fact of any kind – then what seems important from the point of view of someone who will use it is: Can photons be produced on demand? How efficient is it (i.e. how many triggering events lead to a photon in the desired output?)? What is the (spectral) bandwidth? At what rate can photons be produced? It is simply not clear to me that as a user I care about whether its operation depends on some well-defined modes, or otherwise. Furthermore, it is not clear that – setting aside the question of any technological relevance – I will learn something useful about the source. To be sure there are some nice results about how the spatial position of the molecule matters, and its orientation – but the report does not offer a way to control these. Thus, whilst very nice work, it is not clear in what way it offers anything transformation to the field.

Detailed comments

1. Is the Markovian approximation mentioned in the introduction a technological limitation? A comment would suffice...
2. Figure 1. On the face of it there seems little difference between the weak resonator in panel c) and the interrupted waveguide. Is the important thing that the former has a well-defined modal structure, whilst the latter does not?
3. Page 3, paragraph after equation 10. The authors talk of 'the spectrum may be fitted to Eq9'. Surely they mean that Eq9 is fitted to the spectrum (measured)?
4. Page 4, first paragraph, line 3. What is the index difference involved between the gap and wall media? Knowing this would help us understand why the efficiency falls off quickly with gap width.
5. Fig 3. There seems to be a small offset between the peak in panel a) and the dip in panel a), is this important/relevant?
6. In the Supp Info (section 3) the authors talk about voids forming. How big a problem is this? I can imagine this could totally dominate the efficiency.

I. REVIEWER #1

The authors have developed theory in the context of coupling a single photon emitter to a photonic structure. They have developed a theoretical method to deduce the coupling efficiency using only reflection and transmission spectra. They have then applied this theory to an experimental setup in which a DBT molecule is coupled to a SiN waveguide. The broader context for this work is the hot topic of genuine single photon sources integrated with waveguides for quantum information processing.

As the work stands, it chiefly demonstrates that coupling can be inferred from spectra without knowledge of the photonic structure. Their experimental results are consistent with this. The work is perhaps less convincing on the efficacy of their methods as a useful approach to integrate DBT molecules, and I'm not sure whether the field has been advocated in this regard - perhaps the authors could add more of a conclusive statement here.

Our method is indeed a useful approach to integrating DBT molecules in photonic structures. As recommended, we have amplified the text in the discussion section to provide some relevant examples.

II. REVIEWER #2

The manuscript addresses the problem of extracting the coupling efficiency for a single quantum emitter and a waveguide. The emitter in this study was a single dibenzoterrylene (DBT) molecule. The coupling efficiency is the probability for an emitted photon to go into the waveguide mode. The extraction is performed by fitting experimentally measured transmission curves with analytically obtained functions that contain several unknown parameters, including the coupling efficiency. The experimental transmission shows a clear Fano-type line, in agreement with the theoretical expectation.

In general, the subject of the manuscript is quite interesting and is well aligned with the current efforts directed to the practical realization of quantum photonic devices. The manuscript is well written and the results are presented adequately.

While the results seem to be reasonable I do have some concerns about their analysis. I list my comments below. Based on this I feel that the manuscript requires further clarifications and revisions before making any decisions on its publication.

1. The extraction procedure is based on the assumption that the radiative decay rate γ_1 , first defined in Eq. (3), remains unchanged when the molecule is put inside the waveguide. The parameter γ_1 is set using $\gamma_1/\Gamma_1 = 0.33$ based on some measurements for DBT molecules in anthracene from the literature. However, it seems to me that the presence of a waveguide should change the radiative decay rate. The changes may be less significant compared to that for a resonator but still large enough to affect the extraction results. Furthermore, the extraction procedure uses $\tau_1^{-1} = (4.5 \pm 1)ns$. Does this value remain unchanged when the molecules are inserted into the waveguide? The assumptions should be stated more clearly and justified. Furthermore, it makes sense to list in one place all external parameters which are required for the proposed extraction procedure.

We do not assume that γ_1 remains unchanged when the molecule is put inside the waveguide. Just above Eq.(2) we define P_{emit} as the power radiated at frequency ω by the emitter when placed in the structure. We have added the words “with the emitter placed inside the structure” to emphasise that. In Eq.(3) γ_1 is the corresponding radiative decay rate for an emitter inside the structure. The reviewer is right to say that the decay rate of the molecule depends on its environment. We do take that into account when we derive a value of β_{eff} from the measurements because we use the values of γ_1 and Γ_1 measured for DBT in anthracene. Although those values are for bulk anthracene, we believe they apply equally to the molecule in the waveguide and have added new text to justify that. The essential points - both discussed in the Supplementary Information - are that the narrow width of the channel does not increase the measured linewidth, and that the waveguide structure does not significantly alter the branching ratios. The same new text also clarifies that γ_1 and Γ_1 are the only two external parameters needed to deduce a value of β_{eff} from the data. Because there are only two such parameters, we do not feel the need to put them in a table.

2. The extraction procedure uses measurements at several values of the pump power. Is it really necessary? Can one use only weak power to remain in the linear regime? Is there any advantage of going into the nonlinear regime besides being able to verify the power dependence of spectra, which does not seem to be useful for the extraction of the linear parameters.

The reason for taking data at several powers is to extrapolate reliably to the low power limit. This is necessary to establish what amount of power is indeed “low”, and we feel that we would have been rightly criticised if we had not done so. Furthermore, it is a useful practical test of the

theory to make sure that it does describe the higher power case correctly.

3. *The approach yields only some effective coupling coefficient β_{eff} . However, the goal set in the manuscript, line 70, is to find β_{probe} and β_{pump} .*

We have modified the text there and also at the start of the section “Characterisation of the coupling”.

Can the two parameters be obtained directly from the experiments?

Yes they can. We have added text below Eq.11 to elaborate on that.

Does one have to rely further on some geometrical symmetry to obtain β_{probe} and β_{pump} from β_{eff} ?

No. This is explained in the new text below Eq.11. Of course if the geometry is known to be symmetric, then a transmission experiment alone would be sufficient to determine everything.

4. *Page 5, line 268: scanning was performed “over the resonance of a single DBT molecule.” Does that mean that all molecules have non-overlapping spectral lines? Since all molecules seem to be identical, what is the physical mechanism that makes their emission lines to be separated? What is the chance that the lines of several molecules overlap?*

Figure 2(e) is a broad scan that reveals the characteristic Lorentzian resonance peaks of many DBT molecules having a range of resonant frequencies. We have added some words to explain that each molecule has its frequency shifted according to its local environment. Figure 2(f) shows the distribution of resonant wavelengths. We have added a comment to say that the spectral spacing is 0.2 molecule per GHz, so their spectral lines are well resolved.

5. *The extraction is presented only in one particular case, i.e., for a specific molecule. It would certainly be useful to prove the results by applying the procedure to the spectra of several molecules.*

Our present cryostat does not cool to a low enough temperature to give good visibility of the interference feature. This particular molecule was the only one for which we have useful data. It is always nice to have more data, but we feel that this one molecule was sufficient to do the job of testing out our theory.

6. *Page 2, line 89: It says that “the optical reservoir decays faster than all other relevant time scales”. It would be useful to list explicitly the relevant time scales.*

We have made this clearer by rewording the text to say “the local density of electromagnetic states is constant over the spectral linewidth of the emitter”.

7. As far as I understand, while the paper proposes an approach to measure the coupling efficiency it does not specify the location of the molecule for which the efficiency is measured. Is that correct?

We demonstrate an approach to measure the coupling efficiency, which also measures the phase shift of the coupled light. The phase shift does not tell us the position of the emitter, but with the help of numerical simulation it greatly limits the range of positions by reducing the volume to a surface, as described immediately before the Discussion.

8. Figure 3b shows both visibility and asymmetry in percent. However, the numbers quoted in the text use percent only for the visibility parameter ($V_0 = 1.8(1)\%$) but not for the asymmetry parameter $q_0 = 5.2(1) \times 10^{-3}$. It is better to be more consistent.

We now write $q_0 = 0.52(1)\%$.

9. Page 3, line 128: it says “We have also replaced $\overline{4\beta_{\text{pump}}\beta_{\text{probe}}}$ by β_{eff} ”. Why not just write clearly that $\beta_{\text{eff}} = \overline{4\beta_{\text{pump}}\beta_{\text{probe}}}$?

Done.

10. Figs. 3(c,d): the colorbars require some marks and values at the ends.

Done.

11. Page 6, line 333: remove an extra of.

Done.

12. Page 1, line 18: typo in through.

Done.

III. REVIEWER #3

This report concerns the generation of single photons, with a view to applications in quantum information, networks and simulation. The report covers experimental and theoretical aspects, and provides some very nice results. I have a number of comments on detail, points that I imagine the authors will be able to deal with quite easily. They are listed below. Before making those comments I wish to discuss my view about how important this work may be.

First, let me be clear that my own expertise is not a perfect overlap with that of this work. My own research is in nanophotonics, where I do have considerable experience with spontaneous emission into nanostructures.

The experimental work is clear, well executed and interesting. The theoretical aspects are devoted primarily to the problem of how one deals with situations - as presented here - in which the photons produced by the emitter couple to an unknown medley of optical modes (rather than for example a single waveguide mode). It is this novelty/value of this theoretical/conceptual aspect that I am not clear about. If I make a single photon source - such as that used here, or in fact of any kind - then what seems important from the point of view of someone who will use it is: Can photons be produced on demand? How efficient is it (i.e. how many triggering events lead to a photon in the desired output?)? What is the (spectral) bandwidth? At what rate can photons be produced? It is simply not clear to me that as a user I care about whether its operation depends on some well-defined modes, or otherwise.

This is exactly our point. Extinction spectroscopy is a useful technique for determining key properties of a source, and particularly the coupling efficiency, but that used to require a good knowledge of the optical modes. Our technique eliminates this requirement, so extinction spectroscopy can now be applied to any single-photon source, regardless of whether it involves well-defined modes, or not.

Furthermore, it is not clear that - setting aside the question of any technological relevance - I will learn something useful about the source.

Our paper shows how to measure the coupling efficiency, which is a key property of the source, without a tedious (and generally uncertain) measurement of transmission through the optical train. *To be sure there are some nice results about how the spatial position of the molecule matters, and its orientation - but the report does not offer a way to control these. Thus, whilst very nice work, it is not clear in what way it offers anything transformation to the field.*

We do not agree with this. Our method of filling the gap in the waveguide through a microfluidic channel is indeed a way to control the position of the emitter. The density of molecules can be high enough that we are free to choose a molecule that is optimally coupled to the guide and to pick out that specific molecule by the unique frequency of its resonance. Other molecules that are not in the right position or orientation can then be ignored. If necessary the frequency of the chosen molecule can be shifted by and electric fields or by straining the crystal, but those aspects of our research are beyond the scope of this paper, which aims primarily to introduce our new method of measuring the coupling to a general structure. That method is the important primary result of this work.

Detailed comments

1. *Is the Markovian approximation mentioned in the introduction a technological limitation? A comment would suffice. . . .*

No. As we wrote in our original text, “this Markovian approximation is justified for most single-photon sources where a fast optical response is desirable”. Generally a single photon source would aim for high cooperativity while still keeping the memory time of photon bath short compared with the lifetime of the excited emitter. In order to make that clearer, we have re-expressed the approximation as “the local density of electromagnetic states is constant over the spectral linewidth of the emitter”.

2. *Figure 1. On the face of it there seems little difference between the weak resonator in panel c) and the interrupted waveguide. Is the important thing that the former has a well-defined modal structure, whilst the latter does not?*

Yes. We have clarified this by adding a sentence at the end of the text below Eq.(11). “Our method becomes essential when dealing with structures that cannot be described by a simple mode decomposition, such as the waveguide gap depicted in Fig. 1(c).”

3. *Page 3, paragraph after equation 10. The authors talk of ‘the spectrum may be fitted to Eq9’. Surely they mean that Eq9 is fitted to the spectrum (measured)?*

We have corrected this.

4. *Page 4, first paragraph, line 3. What is the index difference involved between the gap and wall media? Knowing this would help us understand why the efficiency falls off quickly with gap width.*

The refractive indices are 1.8 for anthracene, 2.01 for SiN and 1.46 for SiO₂. The efficiency falls off because the field diffracts strongly from the end of the guide, which we could only assess through numerical simulation.

5. *Fig 3. There seems to be a small offset between the peak in panel a) and the dip in panel a), is this important/relevant?*

The peak in Fig. 3(a) is the fluorescence line, which peaks at the resonant frequency. By contrast the dip is the transmission curve, which is a Fano profile. This Fano lineshape does not in general have its dip at the resonant frequency (see also Fig. 3(e)). It is significant that our data conform to the expected Fano profile.

6. *In the Supp Info (section 3) the authors talk about voids forming. How big a problem is this? I*

can imagine this could totally dominate the efficiency.

We do not see any evidence of voids on a microscopic scale, so we are confident that the anthracene is homogeneous within the relevant volume of a single device. The voids are only seen on the large scale shown in Fig. 3(b), and are rare enough that the device yield is high.

Reviewers' Comments:

Reviewer #1:

Remarks to the Author:

The authors have addressed my my main comment by specifying the prospects for full integration of DBT molecules for future quantum information processing. I support the publication of this work in Nature Communications. (Also, apologies to the authors for the typos in my initial review, e.g. advocated -> advanced.)

Reviewer #2:

Remarks to the Author:

see the attached file.

The required details have been added in the revised manuscript. I recommend it for publication. I list a few minor comments below.

Comments:

1. Fig. 3(e): It would certainly be better to explain clearly that the figure shows not the actual transmission, which cannot exceed 100%, but its change in the presence of a molecule. The vertical label should probably be changed.
2. Eq. (16): The values for $\sin \varphi$ and $\cos \varphi$ can be directly obtained from Eqs. (13) and (14) and the corresponding formulas seem to look simpler than Eq. (16). Furthermore, since both \sin and \cos are known there is no need to mention the function $\text{atan2}()$, which is only used in programming.

Reviewer #3:

Remarks to the Author:

I have now looked at the responses made by the authors to the comments made by myself, and those made by the other reviewers. For myself I am happy that the points of issue that I raised have been answered. Whether this work is sufficiently topical for the editorial team at Nature Communications is not something I wish to anticipate, but I do think it now makes a valuable scientific contribution.

I. REVIEWER #2

The required details have been added in the revised manuscript. I recommend it for publication. I list a few minor comments below. Comments: 1. Fig. 3(e): It would certainly be better to explain clearly that the figure shows not the actual transmission, which cannot exceed 100%, but its change in the presence of a molecule. The vertical label should probably be changed.

We have modified the caption to Figures 3(a) and 3(e) to explain that this is the normalised transmission, as defined in Eq. 11.

2. Eq. (16): The values for \sin and \cos can be directly obtained from Eqs. (13) and (14) and the corresponding formulas seem to look simpler than Eq. (16). Furthermore, since both \sin and \cos are known there is no need to mention the function $\text{atan2}()$, which is only used in programming.

We have used the atan2 function because it places τ in the correct quadrant, as we state after Eq. 16. It is true one can determine the signs of \sin and \cos from equations 13 and 14 and use those to establish which quadrant τ lies in, but that seems to us more cumbersome than simply using atan2 . We therefore feel it is best to keep Eq. 16 as it is.